# Determining the rate of infectious disease testing through contagion potential

**Satyaki Roy**[1] *, **Preetom Biswas**[2], **Preetam Ghosh**[3]

**1** Bioinformatics & Computational Science, Frederick National Laboratory for Cancer Research, Frederick, MD, United States of America, **2** School of Computing and Augmented Intelligence, Arizona State University, Tempe, AZ, United States of America, **3** Department of Computer Science, Virginia Commonwealth University, Richmond, VA, United States of America

* satyaki.roy@nih.gov

## Abstract

The emergence of new strains, varying in transmissibility, virulence, and presentation, makes the existing epidemiological statistics an inadequate representation of COVID-19 contagion. Asymptomatic individuals continue to act as carriers for the elderly and immuno-compromised, making the timing and extent of vaccination and testing extremely critical in curbing contagion. In our earlier work, we proposed contagion potential (CP) as a measure of the infectivity of an individual in terms of their contact with other infectious individuals. Here we extend the idea of CP at the level of a geographical region (termed a zone). We estimate CP in a spatiotemporal model based on infection spread through social mixing as well as SIR epidemic model optimization, under varying conditions of virus strains, reinfection, and superspreader events. We perform experiments on the real daily infection dataset at the country level (Italy and Germany) and state level (New York City, USA). Our analysis shows that CP can effectively assess the number of untested (and asymptomatic) infected and inform the necessary testing rates. Finally, we show through simulations that CP can trace the evolution of the infectivity profiles of zones due to the combination of inter-zonal mobility, vaccination policy, and testing rates in real-world scenarios.

## 1. Introduction

Coronavirus disease (COVID-19) is an infectious disease caused by the SARS-CoV-2 virus that has claimed nearly 7 million lives worldwide [1]. Despite the unprecedented strides in vaccination technology, rapid mutations of the virus pose serious threats to human health [2]. These transmissible and virulent strains have been declared the variants of concern (VoCs) by the World Health Organization. Despite the social distancing and vaccination measures, COVID-19 case numbers continue to trickle in. This shows the need for continued efforts to mitigate future outbreaks through pharmaceutical measures (such as vaccines, drugs, etc.,) and nonpharmaceutical measures (public policies, government intervention, mitigation strategies, etc.) [3–5].

**Data Availability Statement:** The data and associated code have been made available on GitHub (https://github.com/satunr/COVID-19/tree/master/CP).

**Funding:** This work was partially supported by NSF CBET-1802588 awarded to P.G. The funders had no role in study design, data collection and analysis, decision to publish, or preparation of the manuscript. P.G. acknowledges salary support from NSF for this work.

**Competing interests:** The authors have declared that no competing interests exist.

Such mitigation measures have been somewhat bolstered by the unprecedented boom in digital technology resulting in the generation of volumes of data and their availability in public repositories [6,7]. Several efforts have been made to leverage this data by devising computational methods for non-pharmaceutical interventions, management strategies, and rehabilitation measures to reduce the spread of transmissible diseases [8,9]. We have witnessed greater collaboration among clinicians, molecular biologists, computer scientists, and applied mathematicians, which opened up several research avenues toward the design of mitigation strategies for future pandemics. The proposed models involve the use of deep machine learning, natural language processing, epidemiology, etc. to determine the socioeconomic, demographic, and clinical factors contributing to the spread of COVID-19 [10–13], plan dynamic lockdowns [14], schedule human mobility [15], etc. It is worth mentioning that similar computational models are being considered to expedite drug discovery and repurposing [16,17].

Despite the rapid strides in computational research, there exists a challenge in curbing an infectious disease, such as COVID-19, spreading at a global scale. The efficacy of non-pharmaceutical interventions is contingent on the accuracy of knowledge of the infected cases. However, a significant fraction of newly infected cases continues to be asymptomatic. In the absence of symptoms, individuals often tend to avoid getting tested, thereby acting as vectors of the disease for the immunocompromised and elderly [18–20]. Despite the availability of mass and rapid testing technology, there is hesitancy to get tested [21,22]. In the broader context, the lack of adequate testing delays the knowledge of an outbreak. Clearly, by the time the number of infected cases reaches a critical mass, the infection has already spread among the public. This raises several questions for policymakers. How does one determine the right time and extent of testing rates? Is there a way to determine trends of contagion from epidemiological statistics alone?

To answer these questions, there have been efforts to incorporate spatiotemporal context into epidemiological studies in order to improve spread predictions. These include the adaptation of existing epidemic model to create age- and gender-based compartmentalization and proposing enforcing distancing measures based on ICU occupancy or combining it with spatial diffusion models to predict contagion [23–25]. These models demonstrate the difference in hospitalization and fatality rates for varying demographic groups. Alongside, age-based stratification, intervention policies are being built on the effect of building types (namely, home, office, school, and mall) [26]. To capture the temporal information in pandemic prediction, computational epidemiologists are considering time-varying model parameters and adding context information such as population flow and meteorological factors, etc. [27–29]. Moreover, inclusion of other factors such as hospital occupancy, emergence of pathogenic mutations to SIR and SEIR epidemic models, and clusters of unvaccinated individuals is considered as a way to predict disease outbreaks, basic reproduction number, mortality rate, and severe infection rate, etc [30,31].

There is little doubt that new mutations of the virus have been a detriment to existing mitigation strategies. Different strains have been associated with massive variations in transmissibility, virulence, and symptoms [32]. The scientific community is divided over the question of having vaccines for different strains as opposed to a universal strain-agnostic COVID-19 vaccine [33]. The heterogeneity in the presentation of the symptoms makes the knowledge of the extent of viral load shedding by asymptomatic carriers an imperative [34]. The research community at large is coping with this uncertainty through the design of epidemiological models that account for the spread effected by the asymptomatic [35], the use of contact tracing mobile applications to track the route of contagion [36–38], and the incentivization of self-quarantine and home-testing kits [39]. These efforts are still hindered by the dearth of knowledge of virus shedding by the carriers and the limitations in the assumptions to model it [40].

In this paper, we explore a metric, termed *contagion potential* (CP), to assess the change in the number of untested (and asymptomatic) individuals in a population, which can help policymakers determine the proper testing rate at any stage in a pandemic. We introduced the concept of CP in our earlier work to gauge the infectivity of a person not in terms of their epidemiological status (i.e., tested infected or not) but as an aggregate of the CP of their recent contacts in a social network [41]. Specifically, a person (shown as a large circle in Fig 1) comes in contact with other individuals (shown as small circles) over time $t = 1,2,3,\cdots$. His low initial CP (close to 0 as shown in green) can assume higher values (close to 1 and colored red) depending on his contact with other high CP individuals. This work attempts to expand the purview of CP to gauge the infectivity of a geographical region, termed zone. Specifically, we consider the following two models characterizing the population-level contagion dynamics: (a) spatial model where individuals inhabit and travel within a zone and influence the CP of the zone by their social mixing; and (b) bulk model where CP is estimated from the

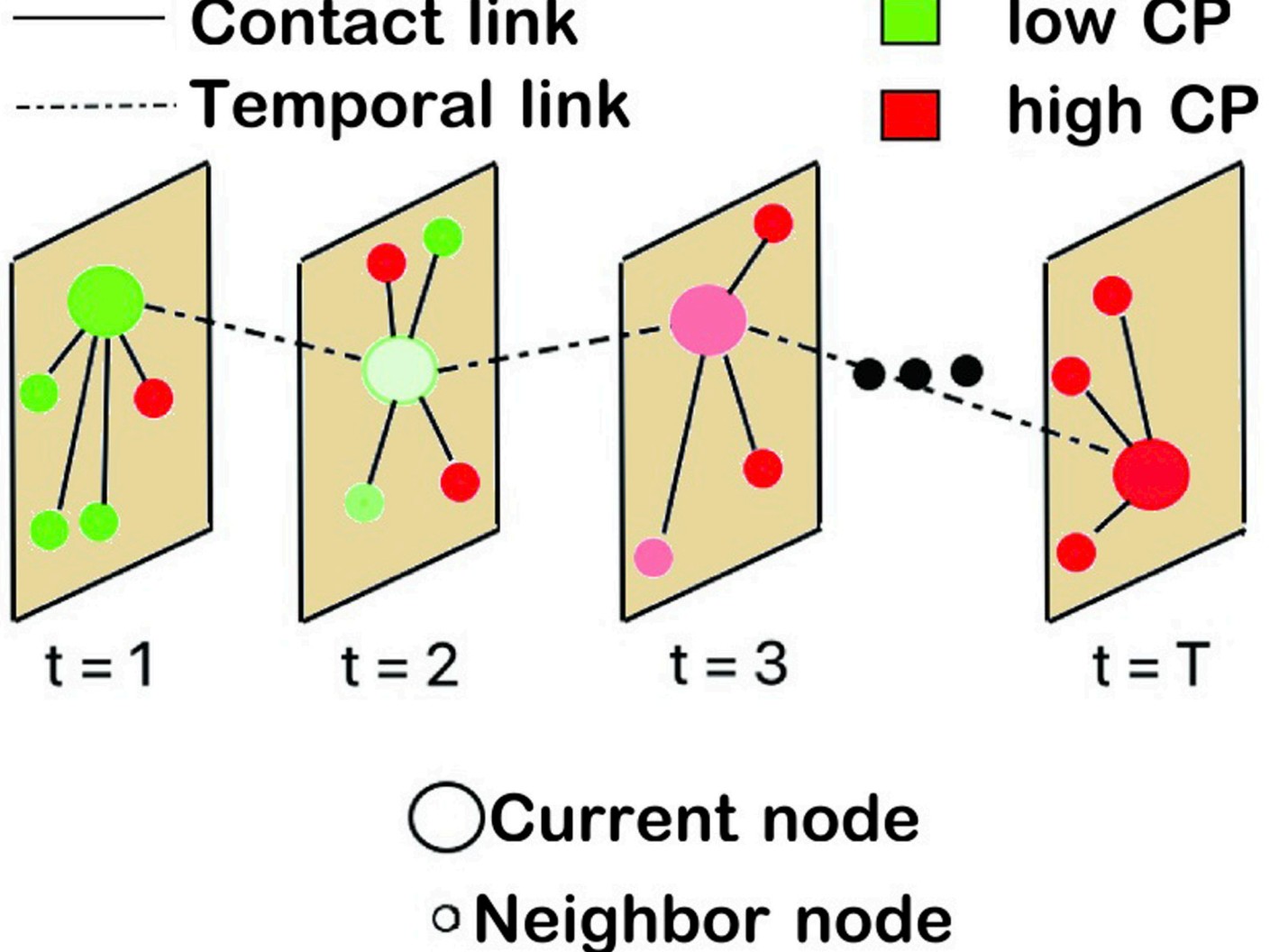

**Fig 1. Evolution of contagion potential (CP) of an individual (shown as a large circle).** Each panel shows the individual's location at a given time. Deep green and red colors indicate low and high CP values, respectively, based on the CP of neighbors (small circle) the individual comes in contact with.

epidemiological data. For the latter, we propose an optimization formulation to estimate mean zonal CP ($\mu$) from the daily infected and recovered numbers. Finally, we present a formulation that captures the evolution in CP due to inter-zonal mobility measured by the traffic flowing into and out of the zones.

We carry out extensive experiments on real and simulated data to demonstrate the efficacy of CP in measuring the infectivity of zones. Given mean CP ($\mu$), zonal population N, and the number of infected I, we derive a metric $\mu N – I$ as a measure of the number of untested (or asymptomatic) infected individuals. Through a spatial SIRS epidemic model, we study the ability of $\mu N – I$ in capturing the true infection trends for varying testing rates, virus strains, reinfection, and superspreader events. Our analysis at a national (as well as state) level of the epidemiological data from Italy, Germany (and New York City, USA) separately elucidates that the temporal change in $\mu N – I$ is a reliable estimator for the absence of testing. Our simulated experiment shows that $\mu N – I$ captures the evolution of the infectivity profiles of zones due to the heterogeneity in (a) inter-zonal mobility and (b) vaccination and testing rates. Moreover, we experimentally elucidate the applicability of CP in a realistic setting of gauging the effect of vaccination policy and contagion on industrial turnover and profit. We conclude the paper with a discussion on the utility of CP in informing public policies on pandemic management and intervention measures.

## 2. Method

### 2.1 SIRS epidemic model

We adopt the Susceptible Infected Recovered Susceptible (SIRS) epidemic model [42]. The susceptible (S) class comprises individuals who are not tested infected. They transition to the infected (I) class with the rate $\beta$ on contact with infected individuals. The infected transition to recovered class at a rate $\gamma$. (Infection rate is the product of basic reproduction number and recovery rate, i.e., $\beta = \gamma \times R_0$ [43]. The recovered individual transitions to S with probability $\delta$. These steps are expressed in terms of ordinary differential equations.

$$\frac{dS(t)}{dt} = -\frac{\beta S(t)I(t)}{N} + \delta R(t) \tag{1}$$

$$\frac{dI(t)}{dt} = \frac{\beta S(t)I(t)}{N} - \gamma I(t) \tag{2}$$

$$\frac{dR(t)}{dt} = \gamma I(t) - \delta R(t) \tag{3}$$

We next consider a spatial variant of the SIRS model, where an individual moves around a geographical region and comes in contact with another individual within a region of radius $r$ units.

**2.1.1 SIRS epidemic model.** We consider a scenario, where individuals are located (and are free to move around) in a geographical region. They come in contact with other individuals within a region of radius $r$ units. The equations below are used to verify the rate of infection spread based on contact (see Fig 2).

*Contact rate.* Consider a population density (given, as per the homogeneous mixing model, by the ratio of the number of individuals and the area of the geographical region) $\rho$. The average number of contacts of an individual at a given time point is a combination of the population density ($\rho$) and the average number of people in the vicinity of the person determined by

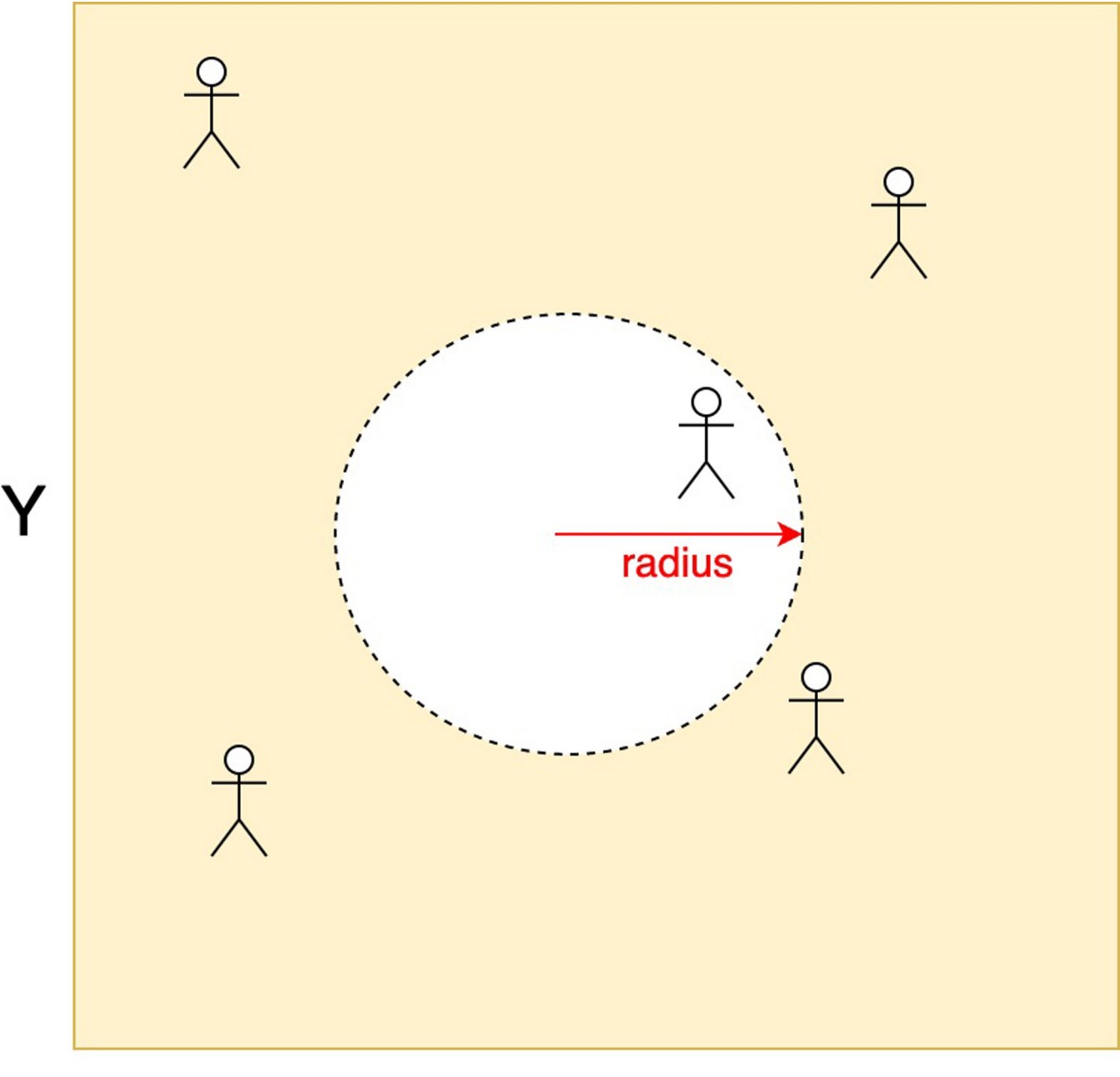

**Fig 2. A geographical region of dimension $X \times Y$ square units, where 5 individuals are located.** The contact area of a person is given by a circle of radius r around their present location.

the circle of radius r (i.e., $\pi \times r^2$) around their current position. Therefore, the contact rate can be calculated as:

$$C = (\pi \times r^2) \times \rho \tag{4}$$

*New infections based on binary infectivity*. Let us assume that the infection propagates by contact between tested infected and recovered individuals. Each person can encounter an infected person with probability given by the ratio of the infected and total population, i.e., $I/N$; they contract the infection at a rate $\beta$ given by the product of the contact rate and infection probability, i.e., $\beta = C \times p$. Thus, the number of new infections is a combination of a number of susceptible S, infected fraction NI and basic reproduction number $R_0$, i.e.,

$$v = \beta \times \frac{I}{N} \times S \tag{5}$$

*New infections based on continuous infectivity*. The scenario discussed in Sec. 2.1.1 considers that the infectivity of a person is binary (i.e., 1 or 0) for an infected and non-infected person, respectively. This makes the mean infectivity $I/N$ (as consider earlier). Now assume that the infectivity, measured by the contagion potential (CP) of a person (see Sec. 2.2), is a continuous value $\in [0,1]$ and the mean CP of the population is given by $\mu$. The number of newly infected is:

$$v = \beta \times \mu \times S \tag{6}$$

**2.1.2 Viral load shedding.** Let the number of days needed to shed the viral load be K. Assuming that the person has the highest contagion potential on day 1, $\mu_1 = 1$ and the CP decay factor is $\eta$. This implies that $\mu_2 = \frac{1}{\eta}$ and $\mu_3 = \frac{1}{\eta^2}$ and $\mu_{t+1} = \frac{1}{\eta^t}$. Since the viral load is shed in $K$ days, let the CP after $K$ days be a small number, say $10^{-10} \approx 0$. Taking logarithms on both sides, we have:

$$\mu_k = \frac{1}{\eta^{K-1}} = 10^{-10} \tag{7}$$

$$\log 1 - (K-1)\log \eta = -10 \tag{8}$$

$$(K-1) \times \log \eta = 10 \tag{9}$$

$$\eta = 10^{\frac{10}{K-1}} \tag{10}$$

**2.1.3 Effect of superspreader events and variants.** Superspreader events are characterized by mass gatherings where people are exposed to the virus due to close proximity with potentially infected individuals. We leverage a class of human mobility models, termed *Human Cell Mobility Model* (HCMM), to model superspreader events [44,45]. As per this model, since each person is part of a social community, they tend to visit places inhabited by the members of their social group. The affinity of person $j$ to visit location (or grid) $i$ is:

$$A(i,j) = \frac{\sum_k M_{j,k}}{|\{k : k \in H_i\}|} \tag{11}$$

In this equation, $k \in H_i$ is a list of people $k$ whose homes are located in grid $i$, while $M_{j,k} \in [0,1]$ is a measure of social association of person $j$ towards person $k$. It is noteworthy to keep the following points in mind.

- As per social network-based models like HCMM, human mobility decisions are guided by the people who are part of their social group. Similarly, superspreader events are large social

gatherings where unvaccinated or immunocompromised people may be exposed to the virus.

- The diagonal elements of $M$ (i.e., $M_{j,j} = 1$) and $M_{j,k} \neq M_{k,j}$ if $i \neq j$.

Another factor influencing the transmissibility and virulence of the virus is its strain. We model the infectivity of different strains by incorporating its basic reproduction number $R_0$ in the rate parameter $\beta$, as $\beta = R_0 \times \gamma$ [43]. As discussed in Sec. 2.1, $\gamma$ is the rate of transition from the infected to the recovered states.

## 2.2 Contagion potential

Recall from Sec. 1, contagion potential (CP) is a measure of the infectivity of an individual, calculated based on the CP of individuals they have come in contact with. Specifically, the CP of any individual μ (with neighbor individuals $v \in N(v)$ at time $t$) at time $t + 1$ is given by:

$$\mu_{t+1}(u) = \alpha \times \mu_t(u) + \beta \sum_{v \in N(u)} \mu_t(v) \qquad (12)$$

We consider the CP of a zone to be summarized by the mean CP of the individuals in a zone. In a practical scenario, there is no social contact network to track social mixing among individuals. Therefore, in order to be applicable in the real world, the CP of zones needs to be calculated from its population-level epidemiological data. To this end, we propose a model that incorporates inter-zonal human mobility into the estimation of CP.

**2.2.1 Calculating zonal contagion potential from bulk population-level data.** Given the infected fraction and recovered fraction of any given zone at time $t$, $v_t$ and $R_t$, its mean contagion potential at $t$ ($\mu_t$) is given by the following optimization formulation:

$$\min_{\beta, \mu_t, R_t} \epsilon \qquad (13)$$

$$s.t. \ \frac{v_t}{u_t \times \beta} + I_t + R_t + \epsilon = 1 \qquad (14)$$

$$0.7 \leq \beta \leq 1.2 \qquad (15)$$

$$1 - I_t \leq R_t \leq 1 \qquad (16)$$

$$I_t \leq \mu_t \leq 1 \qquad (17)$$

Here, the total infected at time t is given by the difference between the sum of new infected and the sum of newly recovered till time t, i.e., $I_t = \sum_0^t v_0^t - \sum_t r_t$. Note that the value of $r_t = I_t \times \gamma$; thus, the value of $\gamma$ in the above optimization is calculated as:

$$\min_{\gamma} \sum \left( \frac{r_t}{I_t} - \gamma \right)^2 \qquad (18)$$

$$s.t. \ 0 \leq \gamma \leq 0.05 \qquad (19)$$

**2.2.2 Incorporating mobility into contagion potential.** We discuss an approach to learn the contagion potential (CP) of zones based on the traffic flowing into and out of them (shown in Fig 3). Consider a row stochastic transition matrix $A$, such that $A_{ij}$ is the probability of

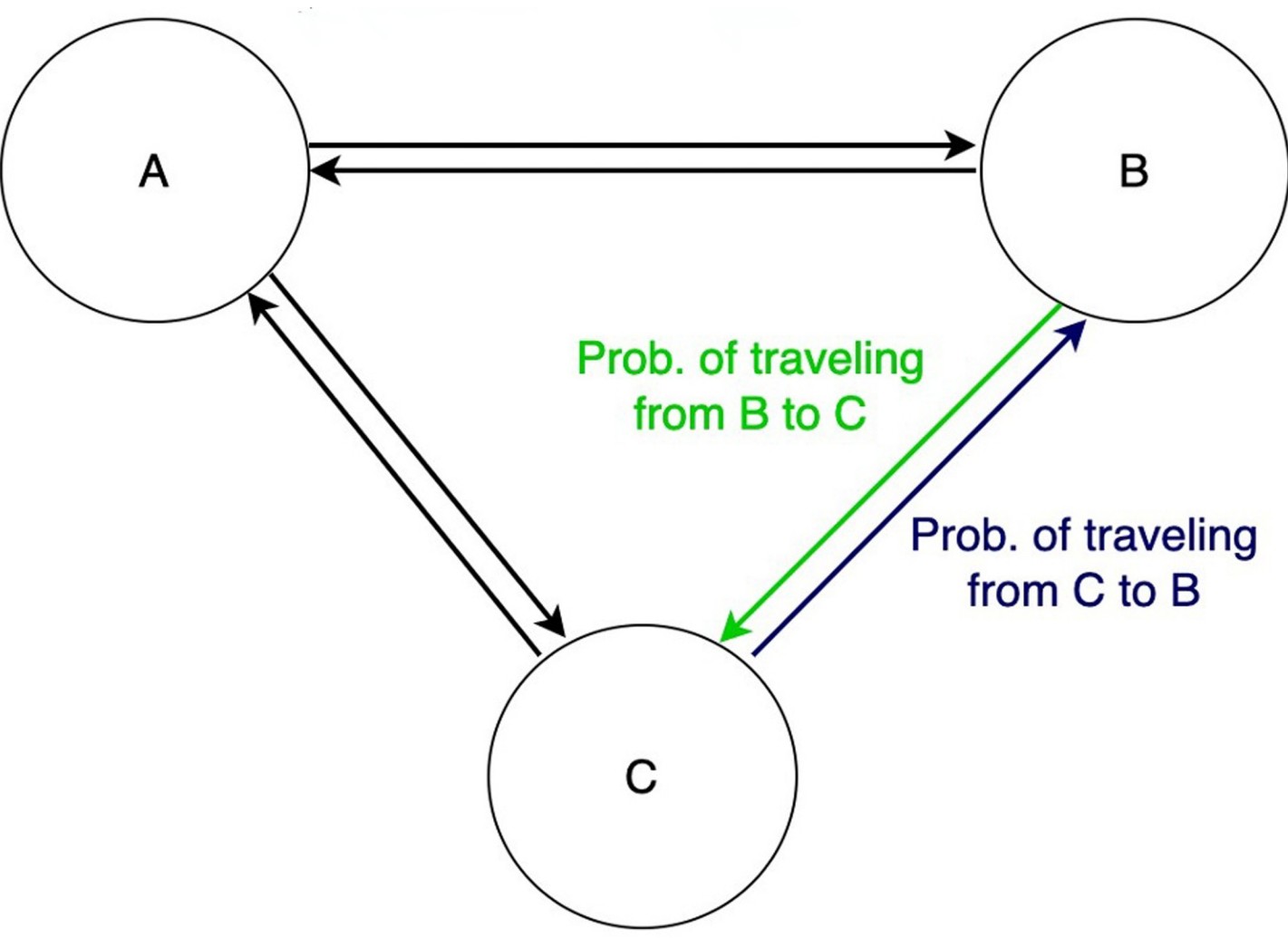

**Fig 3. Three zones A, B, C with in- and out-flow of traffic.**

moving from zone $i$ to $j$. The evolution of CP evolves in two steps: merge and update, as we discuss next. At each day $t$, these steps are invoked once sequentially.

*Merge.* In this step, we estimate the impact of the social mixing of individuals of a given zone i with other zones on the CP of $i$ at the next timepoint $t + 1$. The updated CP at $t + 1$ of zone $i$ is measured as the weighted sum of its own CP ($\mu_i$) and neighbor zone CP ($\mu_j$). The weighted factor is the average number of people entering zone $j$ for $i$, i.e., $N_j \times A_{j,i}$, as shown below.

$$\mu_i^{t+1} = \frac{\mu_i^t \times N_i \times A_{i,i} + \sum_{j \leq n} \mu_i^t \times N_j \times A_{j,i}}{N_i \times A_{i,i} + \sum_{j \leq n} N_j \times A_{j,i}} \tag{20}$$

*Update.* The infectivity of individuals decreases over time, causing the mean CP $\mu$ to undergo decay by a factor $\alpha$. On the other hand, there is an increase in infectivity caused by the social mixing within a zone, given by the rate of disease spread $\beta$. CP of a zone is updated

as follows:

$$\mu_i^{t+1} = (\beta - \alpha) \times \mu_i^t \tag{21}$$

### 2.3. Industrial optimization

We adapt an optimization framework (presented in [46]) in an industrial setting, where industries, varying the type of products they manufacture as well as their vaccination policies, attempt to maximize profit. We consider two types of zones: *industry* and *non-industry* (comprising, residential locations, economic and recreational centers, etc.). An industry zone manufactures products of different types (indexed by *i*), each having different profit margin $\phi_i$, but also requiring different amount of raw material $r_i$ and manpower $\pi_i$. Each individual

- is affiliated to exactly industrial zone,

- travels to and from his industrial workplace and non-industry zones,

- has the expertise to manufacture exactly one type of product, and

- has a vaccination status $V_j = 0,1$ denoting unvaccinated and vaccinated.

We shortlist a subset of zones (i.e., counties) from the traffic flow data of US counties in 2019 (refer to Sec. 3) and estimate normalize the inter-county trip frequency to calculate the transition probability from one zone to another (like in Sec. 2.2.2). The mobility schedule for each individual is a sequence of zones of a fixed length that follows the probability defined in the transition matrix (*A*). Each industry runs the below optimization to maximize profit (see Eq 22), by determining the number of products of type *i*, $n_i$. The production of an industry is constrained on raw materials budget (see Ineq. 23), where a product of type *i* has a raw material cost of $r_i$ Similarly, there is also a budget on manpower (Ineq. 23). The productivity of an individual *j*, $o_j = 1, 0, 0.5$, based on his epidemic status of S, I, and R, respectively. Also, $\omega_{i,j} = 1$ if person *j* he is affiliated to product *i*, and it is 0 otherwise. Therefore, the total cost of manufacturing all products (left-hand side of Ineq. 24) cannot exceed the total productivity of the employees in an industry (right-hand side).

$$\max \sum_i n_i \times \phi_i \tag{22}$$

$$s.t., \ \sum_i n_i \times r_i \leq R^{raw} \tag{23}$$

$$\sum_i (\pi_i \times n_i) \leq \sum_i \sum_j \omega_{i,j} \times o_j \tag{24}$$

The vaccination status $V_j$ of an individual does not feature directly in the optimization formulation. However, $V_j = 1$ (i.e., vaccinated) reduces the probability of transitioning from S to I states by a factor of $(1 - e)$, where $e \in [0,1]$ is the vaccine efficacy.

## 3 Results

We consider three datasets in our experiments. First, we use the daily COVID cases in Germany and Italy from Our World in Data [47] between January 1, 2022—March 31, 2022. In addition to the date and region in the country, the dataset comprises cumulative positive, cumulative deceased, cumulative recovered, currently positive, hospitalized, intensive care, etc. Second, we utilize the daily COVID numbers of New York City Boroughs taken from the NYC Department

of Health and Mental Hygiene [48]. The mentioned datasets are on GitHub (https://github.com/satunr/COVID-19/tree/master/CP). Finally, we also use a realistic human mobility from a dataset taken from mobility-flow dataset in the US during COVID-19 [49]. The experiments were implemented using the Python GeoPy library [50] was employed to connect the zone names with the latitude-longitude coordinates and SimPy library [51] for the spatial model. The SIRS recovery rate parameter ($\gamma$) are inferred from [52]. For reinfection rate $\delta$, we choose a value 0.025 lying in the range given by the reciprocal of the minimum duration (3 months) and maximum duration (2 years) of natural immunity [53]. The infectivity rate $\nu$, reciprocal of decay factor $\eta$ (i.e., $\nu = \frac{1}{\eta}$), is estimated by plugging in viral load shedding period cited in literature [54] of $K = 21$ days into Eq 10. The default parameter values are in Table 1.

### 3.1 Infectivity in a dynamic spatial model

We empirically validate the equations (discussed in Sec. 2.1.1) on the infectivity in a region of $100 \times 100$ square meters. First, we consider the average contact rate observed by varying population density, achieved by varying the population $N = 500, 1000, 1500, 2000$ within the fixed area. Fig 4(A) shows that the observed contact rate in a spatial setting is similar to the rate estimated using Eq 4.

Under the assumption that only the infected people are infectious (i.e., as per binary infectivity in Sec. 2.1.1), we measure the number of new infections for a rate of disease spread $\beta = 0.05, 0.10, 0.15, 0.20$ (given by, $\beta = C \times p$, where contact rate is calculated from Eq 4 for population $N = 500, 1000, 1500, 2000$). Fig 4(B) shows the number of new infections under the binary infectivity assumption to be similar the number of new infections estimated using Eq 5. Next, we calculate the number of new infected under continuous infectivity given by mean CP $\mu$. Once again, the observed new infection count is aligned with the count estimated by Eq 6 (see Fig 4(C)). Finally, we model velocity for 1000 individuals. Each individual moves 50 meters/hour by repositioning themselves to the location (in a randomly chosen direction) on the circumference of a circle of radius 50 meters with the current location as the center of the circle. For $\nu = 100, 150, 200$, an individual makes 2, 3, and 4, such trips, respectively, to model higher social mixing. Fig 4(D) depicts that the average number of contacts increases commensurately with social contact due to the increase in individual velocity.

### 3.2. Testing rate efficacy through contagion potential

To investigate the relationship between testing rates and contagion potential, we consider a scenario where a population of 5000 individuals is placed in an area of $1750 \times 1750$ square meters. The spatial SIRS model (see Sec. 2.1) is invoked with $\gamma = 0.05$ and $\delta = 0.025$. Initially, 5% of the population is infected with CP equal to 1 and the rest are susceptible with CP equal to 0. We carry out a simulation over 60 days with velocity changing every 20 days as follows: 100 meters per hour from day 0 to day 20,550 meters per hour from day 20 to day 40, and 100

**Table 1. Configurable experiment parameters and their default values.**

| Parameter | Value |
|---|---|
| Infection (I) to recovered (R) rate ($\gamma$) | 0.05 |
| Recovered (R) to susceptible (S) rate ($\delta$) | 0.025 |
| Infectivity decay rate (reciprocal decay factor $\nu = \frac{1}{\eta}$) | 0.32 |
| Individual's travel velocity ($\nu$) | 100,550 meters / hour |
| Contact radius ($r$) | 1.82 meters (or 6 feet) |

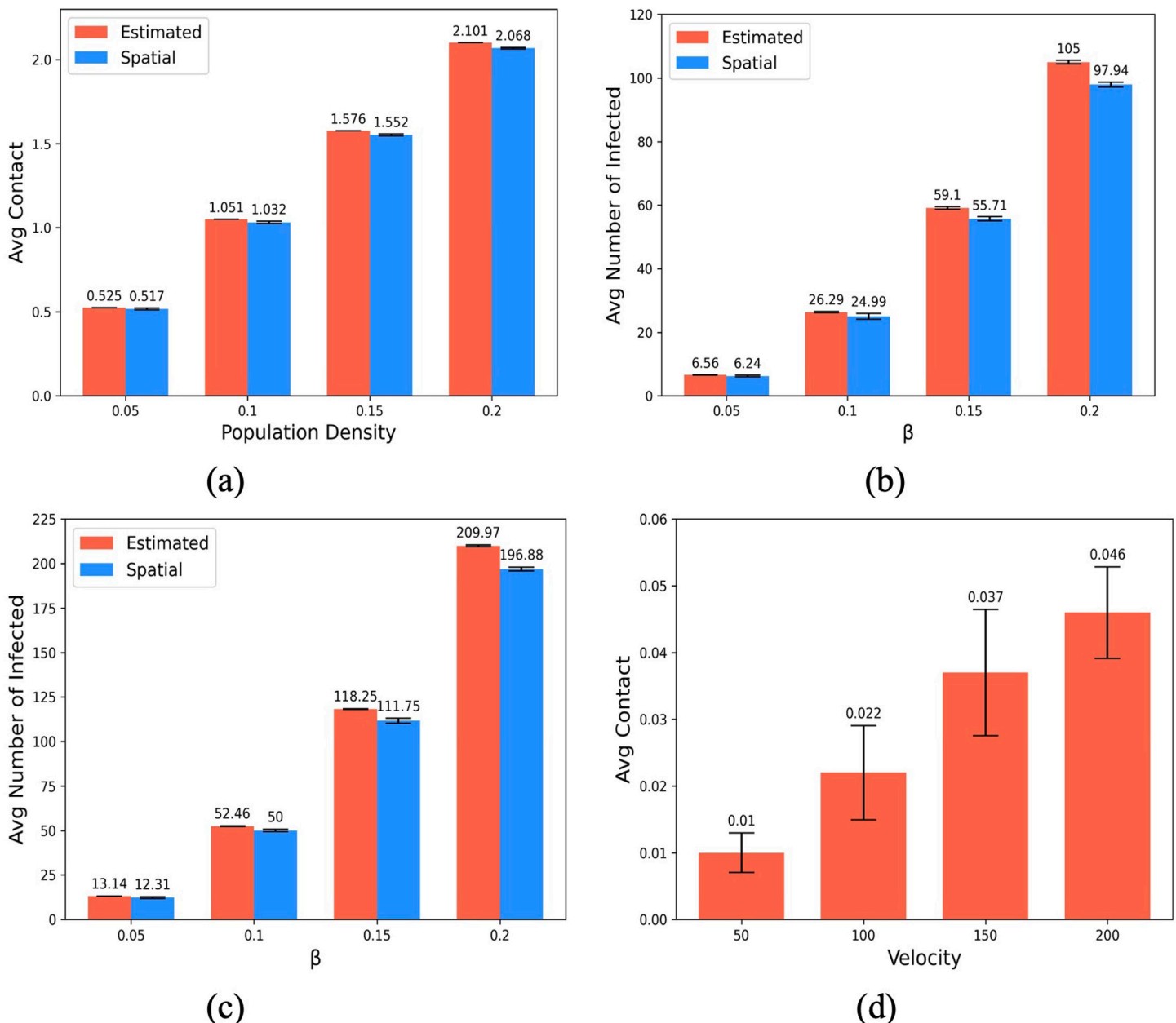

**Fig 4. Infectivity in a Spatial Model.** (a) average contact rate among individuals for varying population density in a spatial model; (b) the number of newly infected for varying rates of disease spread β under the assumption of (b) binary infectivity and (c) continuous infectivity; and (d) increase in average number of contacts with the increase in individual velocity. The error-bars signify a standard deviation from the mean.

meters per hour from day 40 to day 60. As discussed in Sec. 2.1.1 (and shown in Sec. 3.1), the increase in velocity allows for a higher rate of social contacts, resulting in higher contagion.

Fig 5 shows the (1) actual infected, (2) tested infected when 20% of the populated are tested, and (3) tested infected when 50% of the populated are tested in blue dashed, dash-dot and solid lines, respectively. We find that the actual daily infected increase predictably with the increased velocity between day 20 and 40. A testing rate of 20% increase only shows a slight increase in the tested infected numbers, and the tested infected improves for a much higher testing rate of 50%. The metric of $\mu N - I$ is highly sensitive to the increasing contagion for both

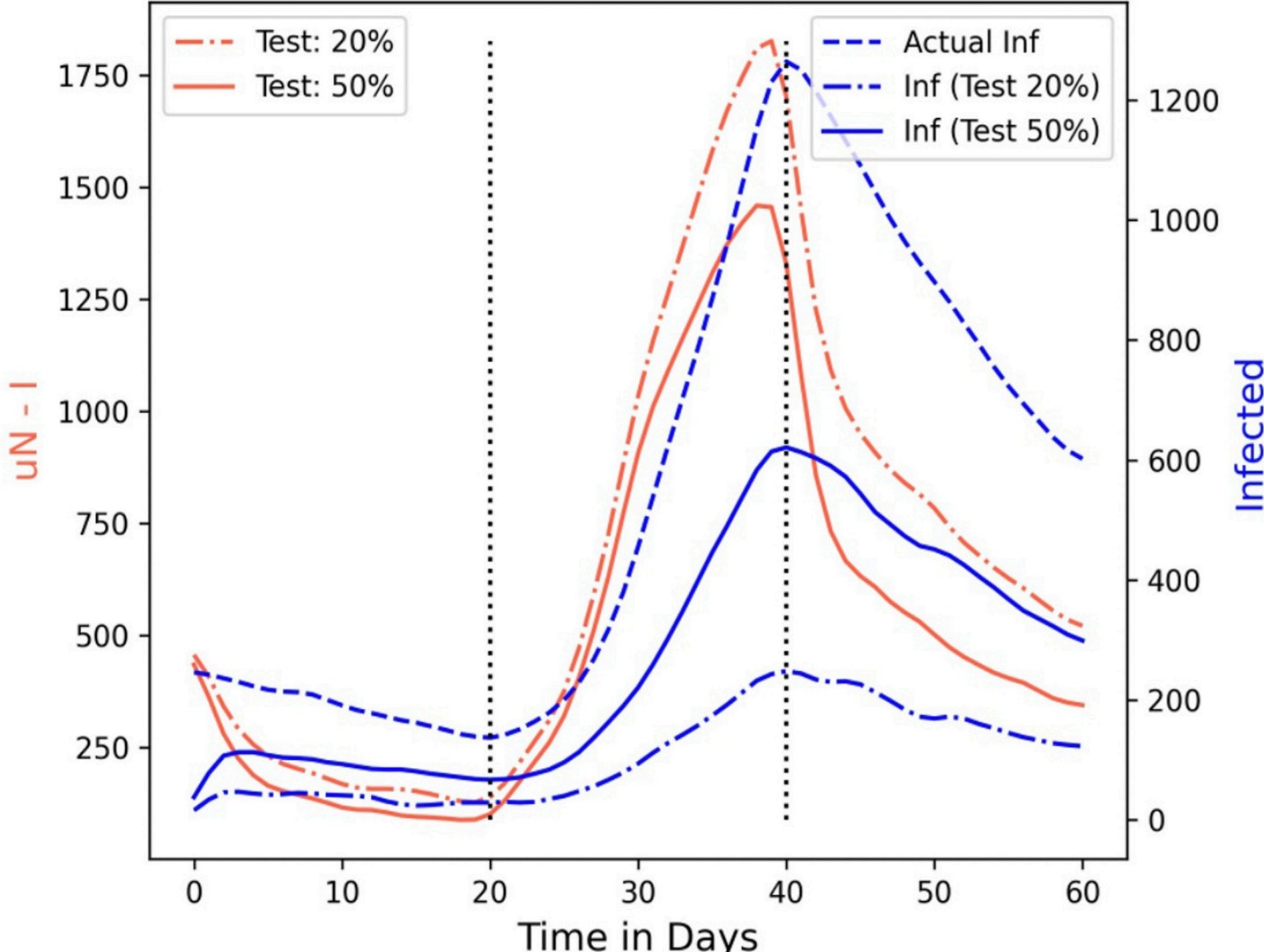

**Fig 5. Comparison of μN − I and tested infected in a simulated environment of 1750 × 1750 square meters and 5000 individuals.** The (1) actual infected, (2) tested infected when 20% of the populated are tested, and (3) tested infected when 50% of the populated are tested in blue dashed, dash-dot, and solid lines, respectively.

testing rates of 20% and 50%. Similarly, the μ$N – I$ value drops very soon after the velocity (and social mixing) falls, elucidating the efficacy of CP in predicting contagion and informing testing rates.

Next, we study whether μ$N – I$ captures the true trends in daily infected under different scenarios, namely, reinfection, virus strain, and superspreader events. To this end, we consider a population of $N$ = 2000 individuals moving across a region of 1200 × 1200 square meters over 60 days. To model superspreaders (through the Human Cell Mobility Model (HCMM) [44,45]), we divide the area into 4 × 4 = 16 grids. We discuss in Sec. 2.1.3 that HCMM causes people to visit locations (or grids) inhabited by people in their social groups (refer to Eq 11)). The level of social attachment of person $i$ towards person $j$ ($i \neq j$) $m_{i,j} \in [0,1]$ is sampled from the standard normal distribution. Given that $H_i$ is a list of individuals whose homes are located in grid $i$, the number of people whose homes are in grid $i$ (i.e., $|H_i|$) is sampled from an

exponential distribution with rate parameter 1. Fig 6(A) and 6(B) show the frequency distribution of the social attachments ($m$) and the populousness of the grids ($H$), both normalized to sum to 1. We also use heatmaps to show the normalized average number of people visiting a given grid when agents are allowed to move randomly (Fig 6(C)) or based on HCMM (Fig 6(D)). Note that, for the latter, grids 11 and 15 are the only two grids to receive a high number of visits commensurate with their $H$ values and are representative of locations hosting major superspreader events. Specifically, each agent makes one move each hour, choosing the next grid with probability given by the HCMM likelihood function, and it is positioned at a random location within the selected grid. We compare HCMM against random mobility, where agents select their next grid location randomly (thereby exhibiting no preference for social ties).

We record the true daily infected, CP $\mu$, $\mu N - I$, and the projected infected numbers (with a testing rate of 20%), for the two values of each of the three settings: (1) virus strain, (2) reinfection rate, and (3) human mobility. Fig 7(A) shows that, for a default setting of the Delta variant (with $R_0 = 3.2$ [55]), reinfection rate $\delta = 0.025$, and random human mobility, the $\mu N - I$ value (denoted by red lines) precedes the peak in true infected numbers (denoted by a blue dashed line). Fig 7(B) shows that the true infected number is marginally lower when we consider no reinfection (i.e., no transition from recovered to susceptible), Delta variant, and random mobility. Similarly, Fig 7(C) shows that the true infected numbers with reinfection, Omicron variant, and random mobility exhibit high contagion than that of the Delta counterpart. This is because the $R_0$ for Omicron is 9.5 [56], making the infected rate $\beta = \gamma \times R_0$ approximately thrice that of Delta. Finally, Fig 7(D) shows that the true infected in the case of superspreader events (with reinfection and Delta) has a lower (yet steadier) peak in comparison with the default random mobility setting.

As depicted in Fig 6(D), the HCMM model causes more visits to grid 11 and 15 by the same group of individuals with friends living in those grids. While this phenomenon leads to less crowding in other zones (causing smaller infection peaks than default), there is a sustained infection trend due to recurring infection among the same group of people traveling to grids 11 and 15. Overall, under all scenarios, $\mu N - I$ captures the real infection trends more accurately than the tested infected curve (blue dotted lines). It is worth noting that Fig 7(A) and 7(C) depict the ability of CP ($\mu$), showed in dash-dotted lines, to capture the effect of $R_0$ on infectivity, as we compare strains with varying $R_0$ keeping other factors (i.e., mobility and reinfection rate) unchanged. The time-evolving value of $\mu$ for Omicron (Fig 7(C)) is higher than that of Delta (Fig 7(A)), accounting for increased infectivity from Omicron ($R_0 = 9.5$) than Delta ($R_0 = 3.2$).

Recall from Sec. 2.2.1 the formulation for estimating the contagion potential (CP) from population-level data, where the learned parameters are rate of disease spread $\beta$, daily recovered numbers $R_t$ and contagion potential $\mu_t$. We apply the optimization to the daily infected COVID-19 numbers in Germany and Italy between January 1—March 1, 2022. Fig 8(A) and 8(B) show the daily susceptible, infected, and recovered numbers (for Germany and Italy, respectively) over a period of 90 days. While the green and blue dotted lines show the actual susceptible and recovered values learned from the optimization, the corresponding solid lines are susceptible and recovered curves smoothed using the Gaussian filter [57] showing the overall contagion trends in the two countries. Fig 9(A) and 9(B) show the daily $\mu N - I$ inferred from the optimization over the same 90-day period and the corresponding daily tested infected numbers for Germany and Italy, respectively. Evidently, $\mu N - I$ spikes when the testing rate is not adequate to capture the rise in contagion, and vice versa.

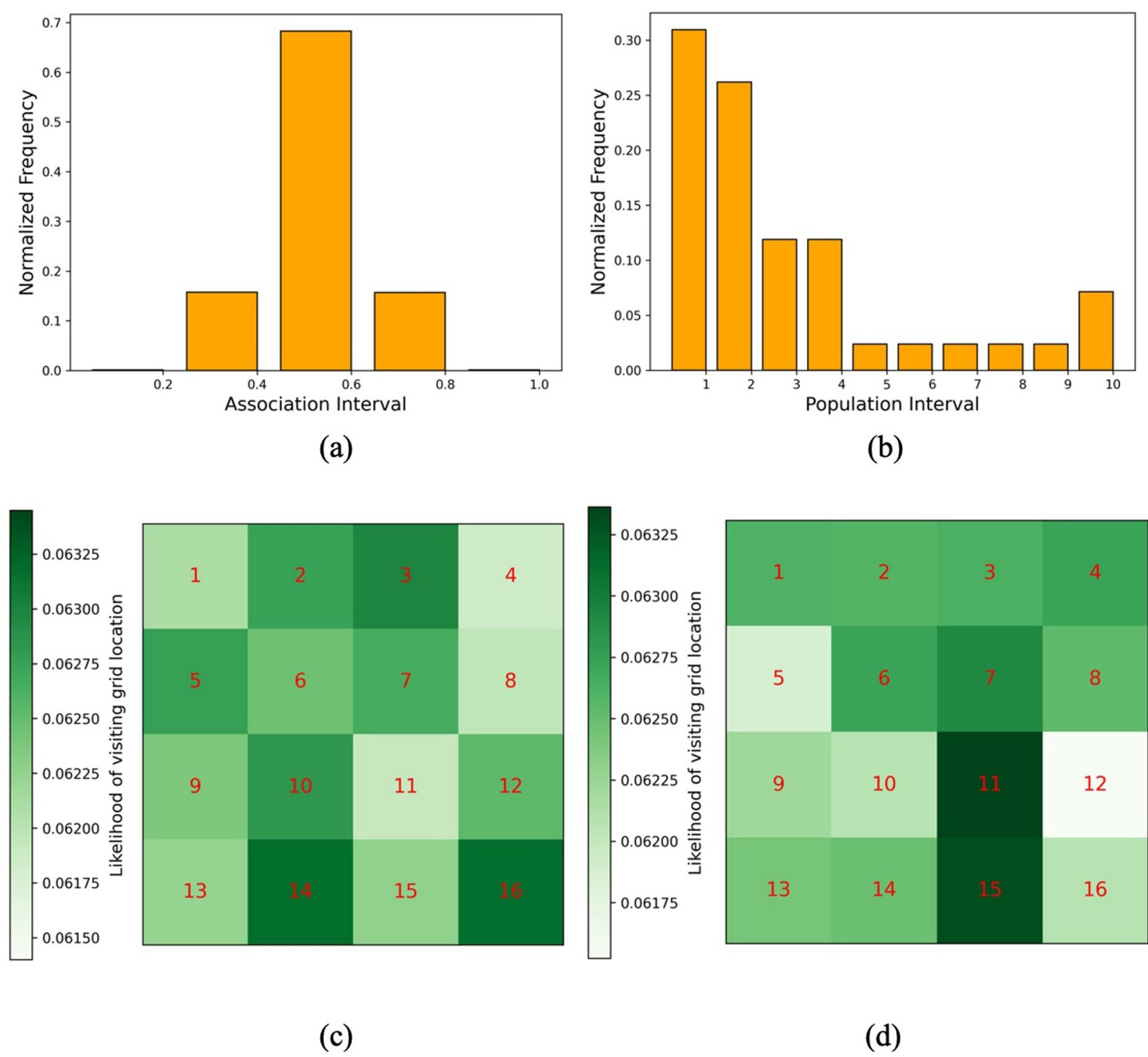

**Fig 6. Effect of superspreader events and viral strains.** Normalized frequency distribution of (a) friendship score m ∈ [0,1] following a standard normal distribution, (b) the number of people whose home is located as a given grid following an exponential distribution with rate parameter 1; heatmap showing the average number of people visiting a given grid when agents follow (c) random mobility and (d) Human Cell Mobility Model [44,45]. the testing rate is not adequate to capture the rise in contagion and vice versa.

### 3.3. Testing rate efficacy through contagion potential

We study whether the effect of inter-zone mobility on the contagion of neighboring zones is captured in their $\mu N - I$ values. We consider 10 zones divided into 2 groups: group 1 (consisting of zones 1–5) and group 2 (consisting of zones 6–10). There is high inter-zonal mobility ($\approx 0.35$) within groups and low inter-zonal mobility ($\approx 0.02$) across groups. (The inter-zonal

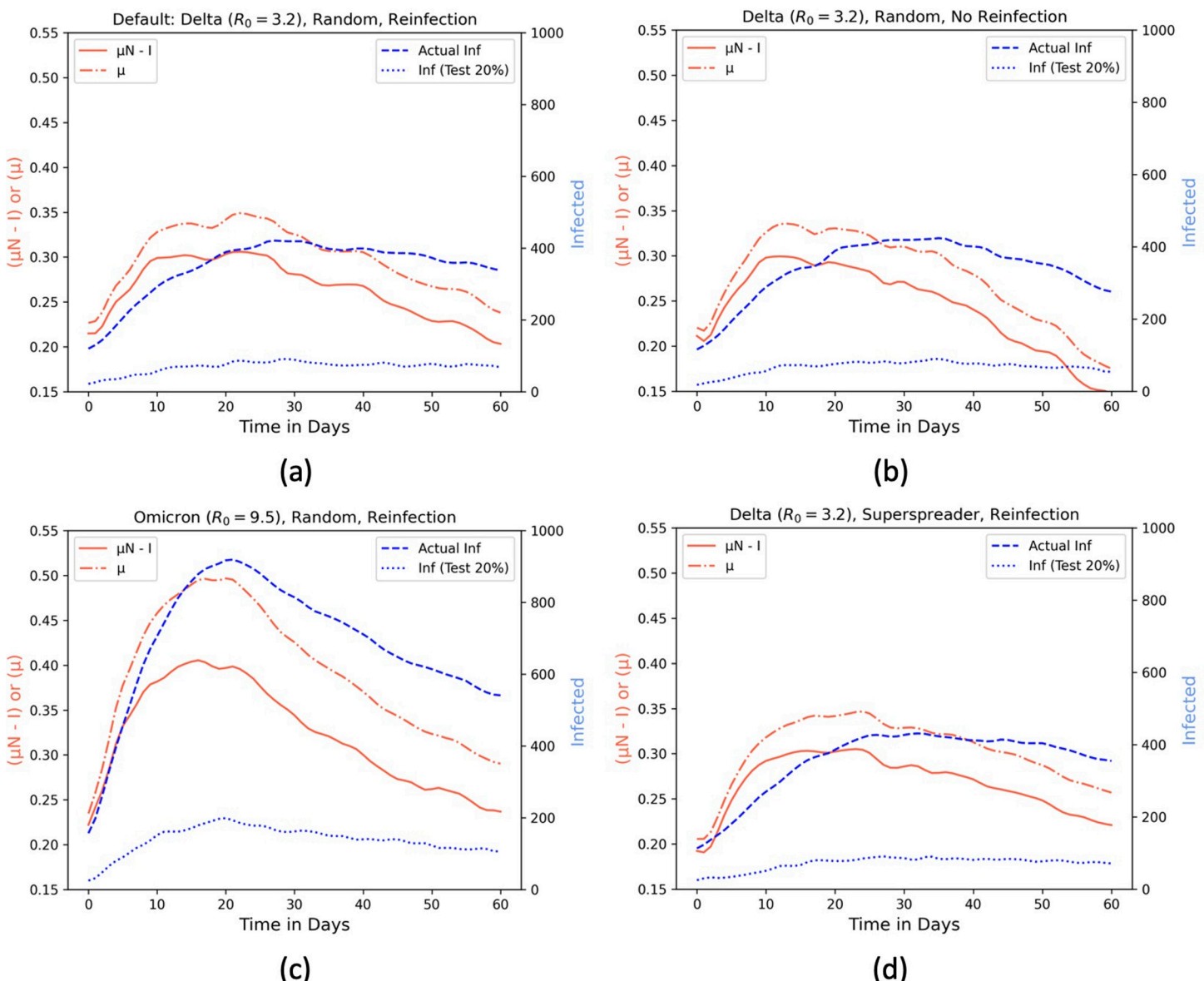

**Fig 7. Effect of superspreader events and viral strains.** Comparison of μ, μN − I and tested infected in a simulated environment of 1200 × 1200 square meters and 2000 individuals based on (a) random mobility, Delta variant, and reinfection (b) random mobility, Delta variant, and no reinfection (c) random mobility, Omicron variant, and no reinfection (d) superspreader, Delta variant, and no reinfection.

mobility of x% from zone a to b implies that $x$% people at zone a at any given time move to zone b.) For an initial population of 1000 individuals per zone with an initially infected fraction of 1%, rate of infection spread $\beta = p \times C = 0.35 \times 0.1 = 0.035$ and duration of 20 days, we record how the change (or slope) in $\mu N - I$ is correlated among zones for varying testing rates.

Consider a scenario where zones have an equal testing rate of 20%. Fig 10(A) shows the heatmap of mean Pearson's correlation of the change in $\mu N - I$ between a pair of zones. We report that the mean correlation within group 1 and group 2 are 0.57 and 0.59, respectively, whereas the mean correlation among zones across groups is 0.43. These correlated changes in $\mu N - I$ within groups stem from high inter-zone mobility. Subsequently, we re-run the experiment for an increased testing rate of zones 2 and 3 to 40%, leaving the testing rate of

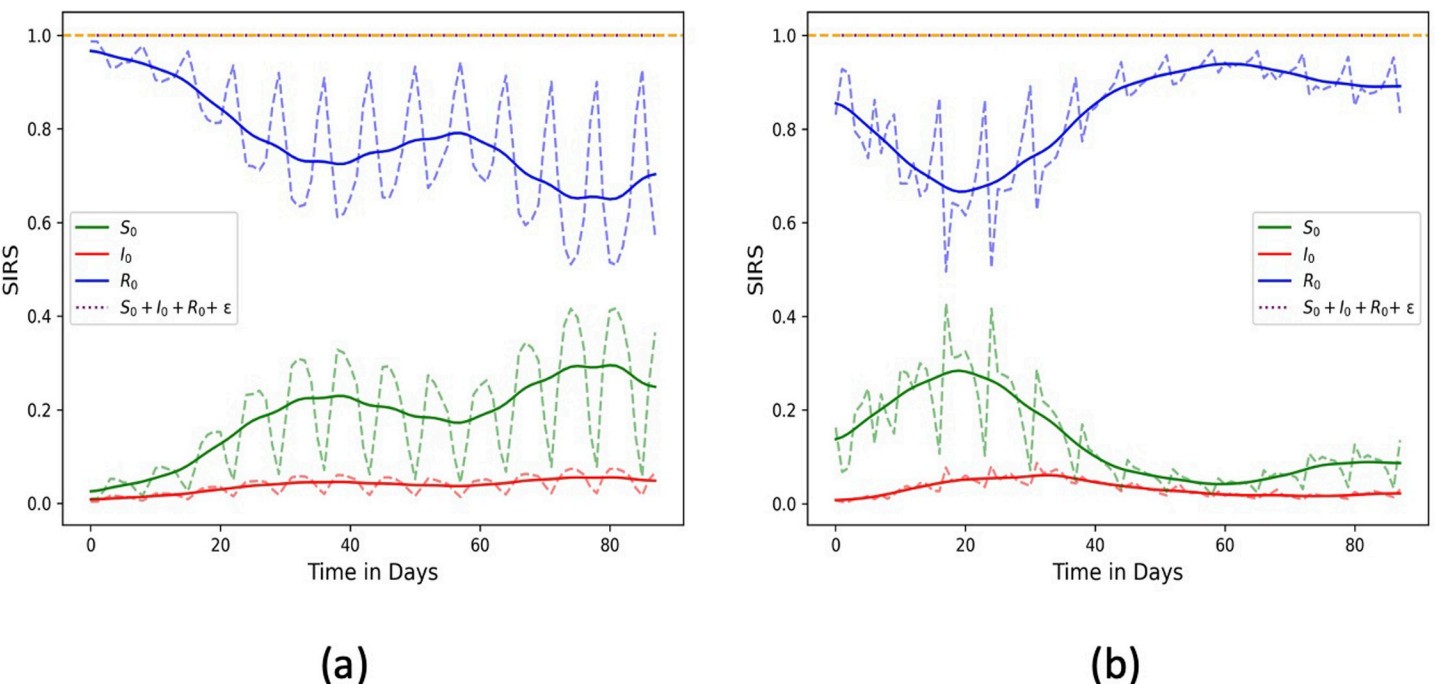

**Fig 8. Daily recovered (and susceptible) fraction of (a) Germany and (b) Italy inferred from the daily infected numbers over a period of 90 days (Jan 1, 2022—Mar 31, 2022).** Green and blue dotted lines show the actual susceptible and recovered values learned from the optimization; the corresponding solid lines are susceptible, and recovered curves smoothed using the Gaussian filter.

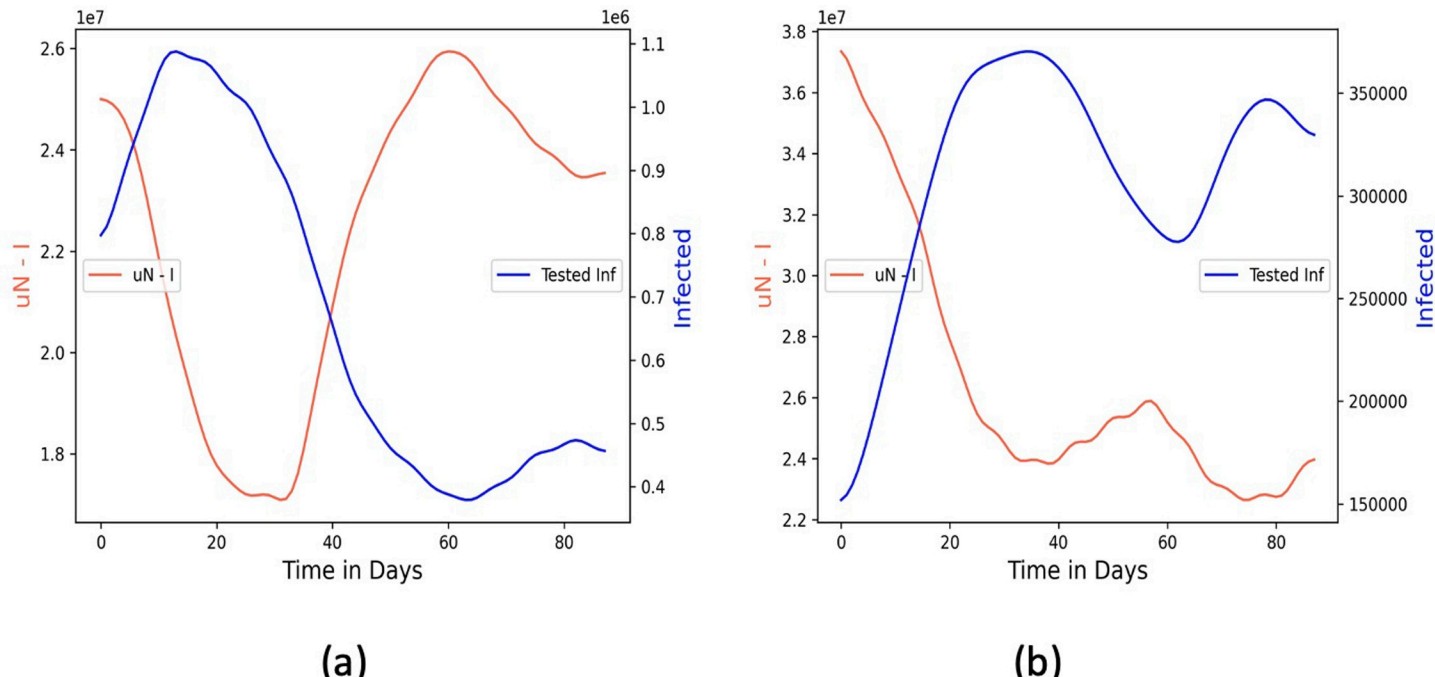

**Fig 9. Opposite trends between μN − I and daily tested infected numbers in (a) Germany and (b) Italy.**

other zones unchanged. Fig 10(B) shows that the correlation within group 1 drops (with a mean of 0.24), depicted as lighter colors in the heatmap. This is because zones 2 and 3, due to higher testing, have reduced the deficit between real and tested infected, and their $\mu N - I$ values no longer correlate with that of zone 1.

### 3.4. Effects of vaccination and testing on contagion

We investigate the daily infection number of the 5 boroughs of New York City between January 1–31, 2022 to understand the effects of vaccination and testing rates on contagion. A high vaccination rate can create herd immunity, offsetting the need for high testing. Together, they influence the number of daily infected and an estimated fraction of untested individuals given by $\frac{\mu N - I}{N}$. To facilitate direct comparison, all numbers are normalized by the population of the respective boroughs (shown in different colors).

Fig 11(A) shows that the normalized $\mu N - I$ of Staten Island is the highest among NYC boroughs, while Brooklyn and Queens have the least normalized $\mu N - I$ over the 30-day period. We delve into the vaccination and testing rates that may explain these numbers. First, Staten Island and Manhattan have the least vaccination rates (Fig 11(B)). Unlike Staten Island, Manhattan, has higher testing rates (Fig 11(C)). Thus, Staten Island has the worst combination of testing and vaccination rates, explaining its highest tested infected fraction (Fig 11(D)) and $\mu N - I$. Brooklyn has a moderate vaccination rate and high testing rate, while Queens balances its low testing rate with a high vaccination rate, resulting in a lower infected fraction and $\mu N - I$. Finally, despite its high vaccination rates, Bronx is placed in the middle in terms of $\mu N - I$ due to low testing.

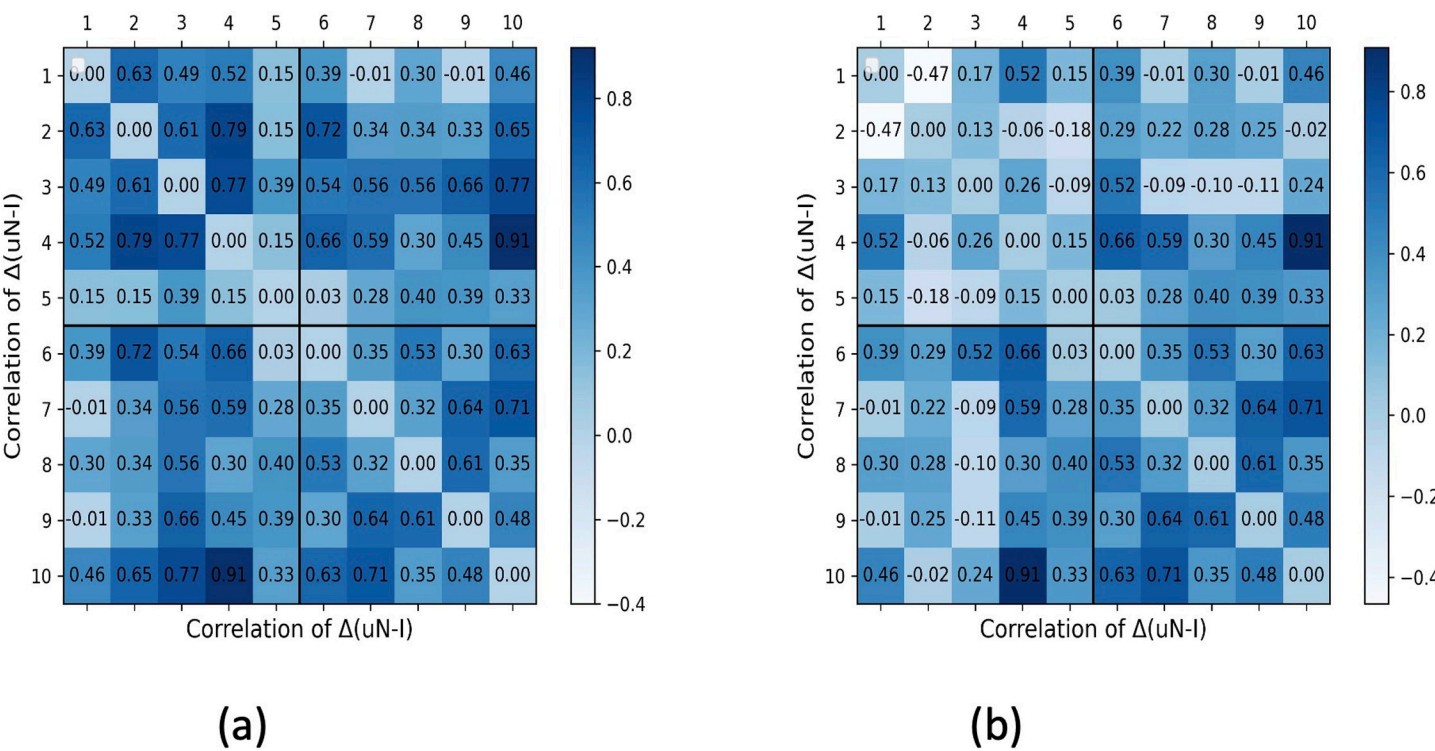

(a)                                               (b)

**Fig 10. Effect of Mobility on CP.** Heatmap of mean Pearson's correlation coefficient of the change in $\mu N - I$ between each pair of zones for (a) equal testing rate and (b) unequal testing rate.

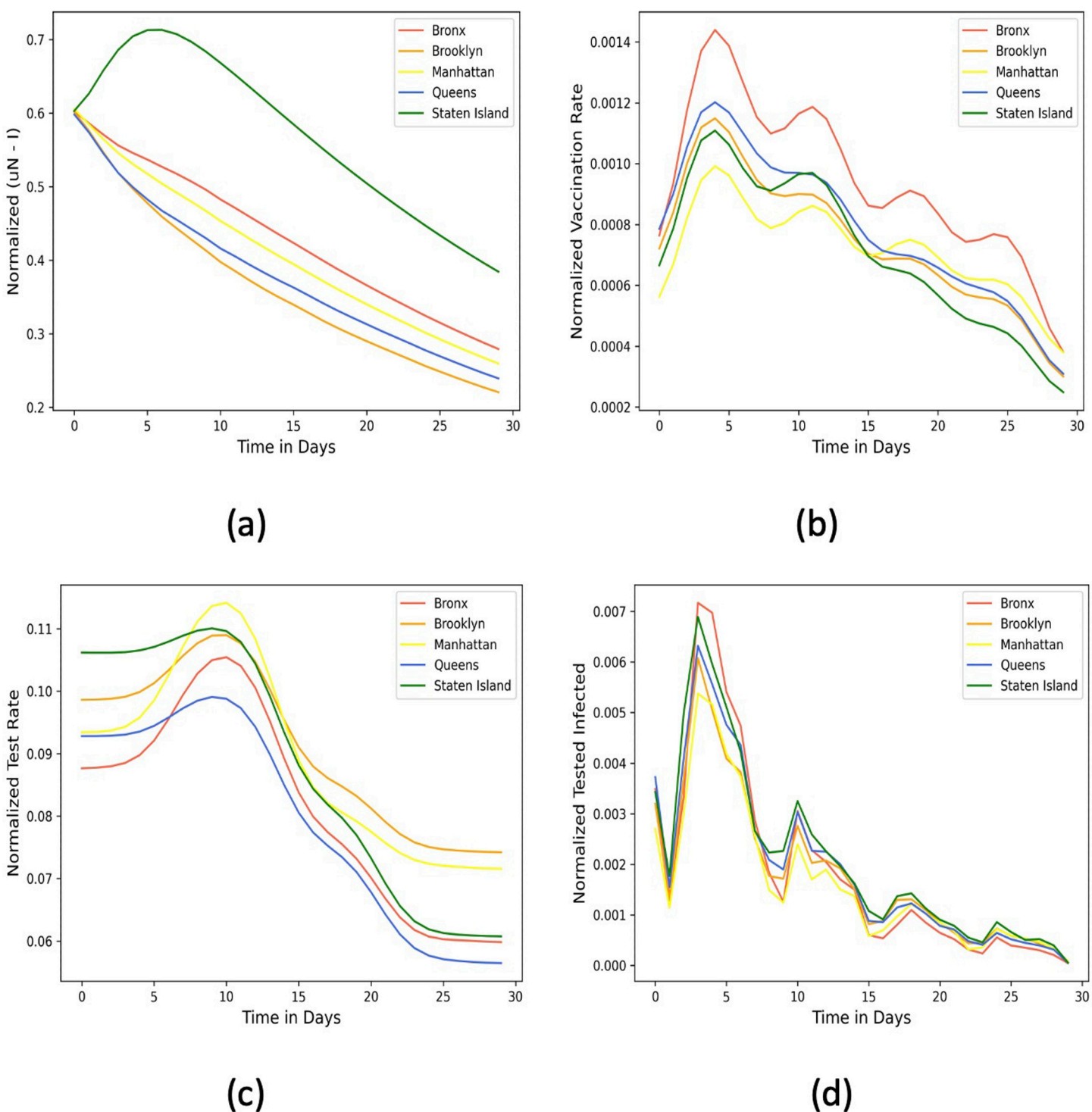

**Fig 11. Effects of Vaccination, Testing on Contagion of the 5 boroughs of New York City.** Normalized (by borough population) (a) vaccination rates; (b) testing rates; (c) infected fraction; and (d) $\mu N - I$.

### 3.5. Effect of contagion on industrial productivity

We design a case study to analyze the effect of contagion, measured in terms of contagion potential (CP), on industrial turnover and profitability (Sec. 2.3). To this end, we consider a

population of 2100 individuals in an urban space of 10 zones, comprising 3 industries and 7 residential zones. As discussed in Sec. 2.3, each individual works on the manufacture of exactly one product at one of the three industries. He follows a mobility schedule that follows a transition probability generated from inter-county mobility dataset in the US (refer to Sec. 1). His schedule is a sequence of zones of 5 zones, containing his industrial workplace and four residential zones. When a person has visited all the zones in the order given by his schedule, he circles back to the first zone.

We consider 2 product types indexed as $i$ = 1,2. Product 1 has a higher manufacture cost, manpower requirement, and profit rate than product 2. Each industry has a raw material budget of $R^{raw}$ = 100000 units and different number of personnel working on both products. The probability of an individual being vaccinated is higher $\sim$ 0.9 if he is working on the more expensive, specialty product (i.e., product 1), and $\sim$ 0.3 otherwise. Vaccinated individuals are less likely to be infected by a factor (1 − $e$), where $e$ = 0.95 is the vaccine efficacy rate. Finally, as pointed out in Sec. 2.3, the manpower budget of an industry is contingent on the epidemic status of its personnel: a susceptible, infected, recovered individual $i$ has productivity $o_j$ = 1,0,0.5, respectively. All parameters for both products are summarized in Table 2.

Fig 12A shows that industry 0 manufactures the highest number of product 1 (as also reported in Table 2), which is the most cost-incurring yet profitable product; industry 2 produces a high number of product 2. Recall from Table 2, individuals manufacturing product 1 are most likely to be vaccinated (with probability 0.9). Since industry 0 has a high proportion of personnel making product 1, their output ($o$) is less likely to suffer from infection, causing their production curves (shown as solid blue and red lines) to be more stable than industries 1 and 2. Similarly, industry 2 with a high proportion of product 2 employees and the least vaccination probability ($\sim$ 0.3) have the highest CP and least profit margin over 60 days (see Fig 12B). Overall, CP captures the effect of vaccination policy of industries on their contagion rates and profitability trends over time.

## 4. Discussions

Our approach explores a metric known as *contagion potential* (CP), which serves as a measure of the infectivity profile within a specific zone. It is important to note that CP does not modify existing epidemic models. Instead, it provides a complementary analysis by quantifying the infectivity of individuals and aggregated to learn the infectivity profile of a geographical region. This unique metric can be applied to any temporal infection data of which the epidemic model can be a potential source. By leveraging CP, we gain valuable insights into the level of infectivity present within a zone, allowing us to better understand the transmission dynamics of infectious diseases. Furthermore, we demonstrate that CP can be effectively estimated using various sources of data, such as social contact information at an individual level (such as contact data) or temporal epidemiological data at a population level, which includes daily infected and recovered statistics. By incorporating CP into our analysis, we enhance our ability to assess the impact of infections and develop more effective strategies to mitigate their spread. The integration of contagion potential (CP) into public policy may change decision-making processes and

**Table 2. Parameter values for product types $i$ = 1,2.**

|  | Product 1 | Product 2 |
|---|---|---|
| Number of individual working on that product type in industries 1, 2, and 3 | (300,200,100) | (400,500,600) |
| Vaccination probability of an individual working on a product type | 0.9 | 0.3 |
| Profit, raw material cost, and manpower cost for a product type | (2,1,2) units | (1,0.4,1) units |

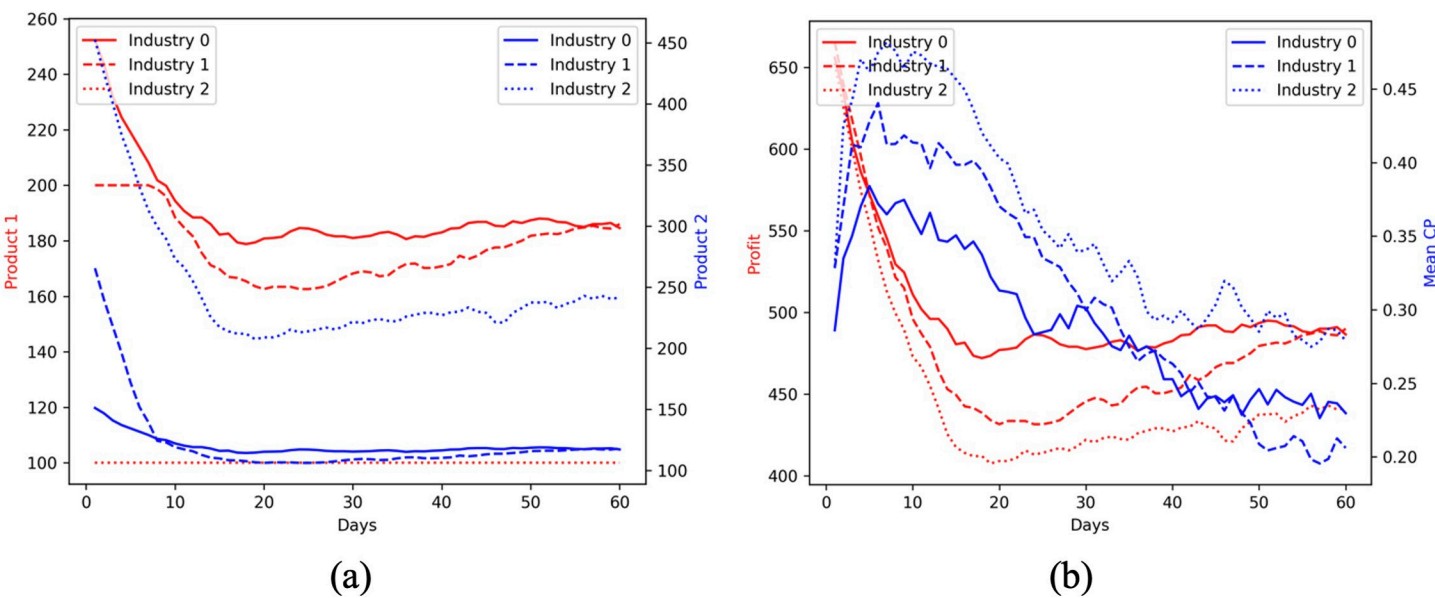

**Fig 12. Effect of contagion on industrial productivity over 60 days.** (a) mean CP and profit earned by industries 1, 2 and 3; (b) number of products of each type $i$ = 1,2 manufactured by each industry.

shape the future of infectious disease control. By leveraging CP as a measure of the infectivity profile within specific geographical zones, policymakers can gain insights into

- targeted and evidence-based intervention strategies,
- the infectivity levels present in different areas, allowing for a more nuanced approach to public health efforts,
- knowledge of high-risk zones and allocate resources, accordingly, focusing on areas with elevated CP to prioritize testing, contact tracing, and targeted interventions,
- effectiveness of earlier measures based on evolving CP, allowing for timely adjustments to control strategies, and
- optimization of resource allocation, minimization of the impact on healthcare systems, and enhancement of overall disease management.

CP can extend beyond infectious diseases, as similar approaches may be applied to other areas such as public safety, disaster response, and environmental health, providing a versatile tool for evidence-based decision-making across domains. As we move into the future, the integration of CP into public policies holds great promise in enabling proactive and effective responses to emerging infectious threats, ultimately safeguarding public health and well-being on a global scale.

While the reproduction number ($R_0$) is a well-established measure of infectivity, the CP metric introduces a distinct and complementary perspective in understanding the dynamics of infectious diseases. While $R_0$ primarily focuses on the intrinsic characteristics of a pathogen, such as its virulence and transmissibility, CP considers the additional influence of social contact resulting from mobility patterns. It recognizes that the spread of diseases is not solely determined by the inherent properties of the pathogen, but also by human behavior and interactions. By incorporating social contact data and considering the movement of individuals within geographical locations, CP captures the effect of mobility and the resultant increased

scope for transmission. This aspect sets CP apart from $R_0$, as it provides a more comprehensive assessment of infectivity by accounting for the interplay between pathogen characteristics and human behavioral factors. We experimentally show in Sec. 3.2 that CP captures the independent effect of varying $R_0$ due to different viral strains on the infectivity profile. Going forward, a combination of such sociodemographic, mobility, and epidemiological factors may enable policymakers and researchers gain insights into the mechanisms driving disease transmission and devise targeted interventions and public health strategies to address the interconnectedness of human interactions and infectious disease dynamics. The goal of our ongoing and future studies will be to

- create simulation and numeric experimental designs to compare how the CP profiles change with differing $R_0$ profiles,

- validate the efficacy of this model using real data and agent-based simulations, and

- incorporate CP into contact-tracing applications to gauge the relative risk of a person in embarking on a trip and schedule their mobility to minimize the risk of contagion.

## Author Contributions

**Conceptualization:** Satyaki Roy, Preetam Ghosh.

**Data curation:** Preetom Biswas.

**Formal analysis:** Preetom Biswas.

**Funding acquisition:** Preetam Ghosh.

**Investigation:** Preetom Biswas.

**Methodology:** Satyaki Roy.

**Software:** Satyaki Roy, Preetom Biswas.

**Supervision:** Preetam Ghosh.

**Validation:** Satyaki Roy, Preetom Biswas, Preetam Ghosh.

**Visualization:** Preetom Biswas, Preetam Ghosh.

**Writing – original draft:** Satyaki Roy, Preetom Biswas, Preetam Ghosh.

**Writing – review & editing:** Satyaki Roy, Preetom Biswas, Preetam Ghosh.

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
