## [Decision Letter · Decision Letter 0]

13 Jan 2023

PGPH-D-22-01697

Determining the Rate of Infectious Disease Testing through Contagion Potential

Dear Dr. Roy,

Thank you for submitting your manuscript to PLOS Global Public Health. After careful consideration, we feel that it has merit but does not fully meet PLOS Global Public Health’s publication criteria as it currently stands. Therefore, we invite you to submit a revised version of the manuscript that addresses the points raised during the review process.

Please note that we have only been able to secure a single reviewer to assess your manuscript. We are issuing a decision on your manuscript at this point to prevent further delays in the evaluation of your manuscript. Please be aware that the editor who handles your revised manuscript might find it necessary to invite additional reviewers to assess this work once the revised manuscript is submitted. However, we will aim to proceed on the basis of this single review if possible. 

The reviewers have recommended further discussion on certain confounding variables, such as the rate of infection and the prevalence of super spreaders and chronically infected individuals, and how they may have affected the results. Their comments can be found below

We look forward to receiving your revised manuscript.

Kind regards,

Lucinda Shen, MSc

Staff Editor

Journal Requirements:

2. We ask that a manuscript source file is provided at Revision. Please upload your manuscript file as a .doc, .docx, .rtf or .tex.

3. Please provide separate figure files in .tif or .eps format only and remove any figures embedded in your manuscript file. Please also ensure that all files are under our size limit of 10MB.

Additional Editor Comments (if provided):

Reviewers' comments:

Reviewer's Responses to Questions

**Comments to the Author**

1. Does this manuscript meet PLOS Global Public Health’s publication criteria? Is the manuscript technically sound, and do the data support the conclusions? The manuscript must describe methodologically and ethically rigorous research with conclusions that are appropriately drawn based on the data presented.

Reviewer #1: Yes

2. Has the statistical analysis been performed appropriately and rigorously?

Reviewer #1: Yes

3. Have the authors made all data underlying the findings in their manuscript fully available (please refer to the Data Availability Statement at the start of the manuscript PDF file)?

Reviewer #1: Yes

4. Is the manuscript presented in an intelligible fashion and written in standard English?

Reviewer #1: Yes

5. Review Comments to the Author

Reviewer #1: The authors propose to extend the concept of Contagion Potential (CP) to assess the dynamics of COVID-19 transmission, using spatial and SIR models. The concepts, assumptions and equations are coherent, and the inclusion of mobility measures, as well as vaccination and testing, as factors influencing the dynamics of SARS-Cov-2 transmission is appropriate. Results are compelling, especially for contributing to public policies in a controversial topic, i.e., the importance of lockdown or other nonpharmaceutical measures. There are few co-factors that in my opinion should be included (or at least discussed as possible limitations), mainly the differential Ro of variants, the relevance of reinfections (thus contradicting a typical SIR model), the prevalence of superspreaders and chronically infected individuals.

6. PLOS authors have the option to publish the peer review history of their article (what does this mean?). If published, this will include your full peer review and any attached files.

**Do you want your identity to be public for this peer review?** For information about this choice, including consent withdrawal, please see our Privacy Policy.

Reviewer #1: No

---

## [Decision Letter · Decision Letter 1]

11 May 2023

PGPH-D-22-01697R1

Determining the Rate of Infectious Disease Testing through Contagion Potential

Dear Dr. Roy,

Thank you for submitting your manuscript to PLOS Global Public Health. After careful consideration, we feel that it has merit but does not fully meet PLOS Global Public Health’s publication criteria as it currently stands. Therefore, we invite you to submit a revised version of the manuscript that addresses the points raised during the review process.

The manuscript has been evaluated by two reviewers, and their comments are available below.

Although reviewer 2 is satisfied with the manuscript, reviewer 3 has several requests for additions and clarifications, including a more thorough contextualization of the research. 

Could you please carefully revise the manuscript to address all comments raised?

We look forward to receiving your revised manuscript.

Kind regards,

Steve Zimmerman, PhD

PLOS Staff Editor

Journal Requirements:

2. Please provide separate figure files in .tif or .eps format only and remove any figures embedded in your manuscript file. Please also ensure all files are under our size limit of 10MB.

Additional Editor Comments (if provided):

Reviewers' comments:

Reviewer's Responses to Questions

**Comments to the Author**

1. If the authors have adequately addressed your comments raised in a previous round of review and you feel that this manuscript is now acceptable for publication, you may indicate that here to bypass the “Comments to the Author” section, enter your conflict of interest statement in the “Confidential to Editor” section, and submit your "Accept" recommendation.

Reviewer #2: All comments have been addressed

Reviewer #3: All comments have been addressed

2. Does this manuscript meet PLOS Global Public Health’s publication criteria? Is the manuscript technically sound, and do the data support the conclusions? The manuscript must describe methodologically and ethically rigorous research with conclusions that are appropriately drawn based on the data presented.

Reviewer #2: Yes

Reviewer #3: Yes

3. Has the statistical analysis been performed appropriately and rigorously?

Reviewer #2: Yes

Reviewer #3: Yes

4. Have the authors made all data underlying the findings in their manuscript fully available (please refer to the Data Availability Statement at the start of the manuscript PDF file)?

Reviewer #2: Yes

Reviewer #3: Yes

5. Is the manuscript presented in an intelligible fashion and written in standard English?

Reviewer #2: Yes

Reviewer #3: Yes

6. Review Comments to the Author

Reviewer #2: (No Response)

Reviewer #3: Since this is the second round of this paper, I will skip the summary part of the review.

Overall, the study is intresting but I have several issues in its current form:

Major:

- The authors are focusing on a very complex and well studied subject while not providing a sepearated Related Work section.

I do not say they have to, and can put it as part of the Introduction section but must provide to the user a more detailed review of the subject.

As far as I see it, the authors should review and explain how their work is different in several fronts:

A. Graph-based spatio temporal SIR models

B. Multi-strain and multi-mutation SIR models

C. The social compoenent in pandemic spread and how other modeled it.

I suggest the authors to review and maybe inlucde the following, alongside other works:

1. doi: 10.1109/ACCESS.2022.3149956.

2. https://doi.org/10.1371/journal.pone.0246961

3. https://doi.org/10.1016/j.aml.2020.106617

4. https://doi.org/10.1016/j.cnsns.2021.106176

5. doi: 10.3934/mbe.2020372

6. https://doi.org/10.1016/j.idm.2021.01.006

7. https://doi.org/10.1038/s41598-021-94609-3

- The mobility and social mixing approach is intresting but should be better supported with real data.

For example, https://www.sciencedirect.com/science/article/pii/S0038012123000538 used graph-based epidemiological model where individuals move between nodes according to logic and the nodes are different.

They support it by historical records. I suggest the authors would do the same.

- The model's limitations and applications should be discussed better.

Minor:

- How does the values in Table 3 are obtained?

- Explain the bars in Fig. 4 - I guess they are one STD but this is not clear from the caption.

- The design of fig 6 color-wise is strange.

Compliments:

- The abstract is well written.

- Fig 1 is very cute.

7. PLOS authors have the option to publish the peer review history of their article (what does this mean?). If published, this will include your full peer review and any attached files.

**Do you want your identity to be public for this peer review?** For information about this choice, including consent withdrawal, please see our Privacy Policy.

Reviewer #2: **Yes: **Sevda Molani

Reviewer #3: **Yes: **Teddy Lazebnik

---

## [Decision Letter · Decision Letter 2]

23 Jun 2023

PGPH-D-22-01697R2

Determining the Rate of Infectious Disease Testing through Contagion Potential

Dear Dr. Roy,

Thank you for submitting your manuscript to PLOS Global Public Health. After careful consideration, we feel that it has merit but does not fully meet PLOS Global Public Health’s publication criteria as it currently stands. Therefore, we invite you to submit a revised version of the manuscript that addresses the points raised during the review process.

We look forward to receiving your revised manuscript.

Kind regards,

Abram L. Wagner, PhD, MPH

Academic Editor

Journal Requirements:

2. 

Additional Editor Comments (if provided):

Several peer reviews have already reviewed the paper and I believe the authors have responded well to their comments on the methods and results. I do think the discussion needs to be fleshed out a bit more. I would recommend having a (brief) paragraph discussing how this relates to any other previous work.

It also could be helpful to explain out a bit more what the implications of the contagion potential are, especially to a more general audience who may not be as familiar with mathematical modeling.

Is there a way to distinguish it colloquially from R_0. Also, could contagion potential be used to explain epidemic potential at different geographical levels? (e.g., Masters et al. http://www.pnas.org/lookup/doi/10.1073/pnas.2011529117)

Reviewers' comments:

Reviewer's Responses to Questions

**Comments to the Author**

1. If the authors have adequately addressed your comments raised in a previous round of review and you feel that this manuscript is now acceptable for publication, you may indicate that here to bypass the “Comments to the Author” section, enter your conflict of interest statement in the “Confidential to Editor” section, and submit your "Accept" recommendation.

Reviewer #3: All comments have been addressed

2. Does this manuscript meet PLOS Global Public Health’s publication criteria? Is the manuscript technically sound, and do the data support the conclusions? The manuscript must describe methodologically and ethically rigorous research with conclusions that are appropriately drawn based on the data presented.

Reviewer #3: Yes

3. Has the statistical analysis been performed appropriately and rigorously?

Reviewer #3: Yes

4. Have the authors made all data underlying the findings in their manuscript fully available (please refer to the Data Availability Statement at the start of the manuscript PDF file)?

Reviewer #3: No

5. Is the manuscript presented in an intelligible fashion and written in standard English?

Reviewer #3: Yes

6. Review Comments to the Author

Reviewer #3: The authors addressed all my comments

7. PLOS authors have the option to publish the peer review history of their article (what does this mean?). If published, this will include your full peer review and any attached files.

**Do you want your identity to be public for this peer review?** For information about this choice, including consent withdrawal, please see our Privacy Policy.

Reviewer #3: **Yes: **Teddy Lazebnik

---

## [Editor Report · Decision Letter 3]

10 Jul 2023

Determining the Rate of Infectious Disease Testing through Contagion Potential

PGPH-D-22-01697R3

Dear Dr. Roy,

We are pleased to inform you that your manuscript 'Determining the Rate of Infectious Disease Testing through Contagion Potential' has been provisionally accepted for publication in PLOS Global Public Health.

Best regards,

Abram L. Wagner, PhD, MPH

Academic Editor